# Assessing the Core Variables of Business Managers’ Intuitive Decision Ability: A Review and Analysis

**DOI:** 10.3390/bs12110409

**Published:** 2022-10-24

**Authors:** Peter L. Nuthall

**Affiliations:** Department of Land Management and Systems, Lincoln University, Lincoln 7647, New Zealand; peter.nuthall@lincoln.ac.nz

**Keywords:** business decisions, ‘objectively informed intuition’, modelling business intuition, decision skill, evolution of intuition

## Abstract

Business decisions are frequently based on informed intuition in contrast to a formal analysis. Early man used simple intuition, but through time knowledge increases allowed decision makers (DMs) to move to ‘objectively informed intuition’ (OII). This uses inherent and learnt cognition at both unconscious and conscious levels. A model of business OII is proposed and evaluated using as variables the managers’ personal characteristics and their unique set of objectives. The resultant equation allows assessing decision quality and provides a framework for DMs to work on improvements relative to their objectives. The literature suggests OII stems from a DM’s makeup (business related phenotype), training and experience in a dynamic trio leading to the defining equation. Analyses show business related phenotype is the most important determinant as well as confirming the proposed theory on the determinants of OII success. Practical methods of improving OII are reviewed, and issues worth further investigation outlined. This research is the first encompassing quantitative relationships explaining business OII quality thus enabling improving OII. Suggested further research may refine the equation and expand its core base. This work involves a range of disciplines as different aspects of human characteristics impact on how decisions are made.

## 1. Introduction

Decisions involve observation, cognitive processing and subsequent action based on mentally assessed conclusions [1]. This process is referred to as intuition when it is largely subconscious and rapid. As the majority of decisions involve both a component of simple intuition together with a varying level of conscious cognition, it is important to understand the process to enable improving decisions. This paper contains reviews, proposals and issues related to the most commonly used time involving form of business decision intuition referred to here as ‘objectively informed intuition’ (OII). Most importantly, the reviews lead to developing an equation defining the core parameters determining business OII quality. This has never been achieved before and provides a major contribution to improving decisions as well as a basis for further research. 

Appelt et al. [2] introduce decision making by asking (p. 252) ‘How much of human behaviour is due to the ‘person’ versus the ‘situation’? They note three factors impinge on human decision making … decision features, situational factors and individual differences. Salas et al. [3], McElroy et al. [4], and others, came to a similar conclusion. While all these factors impact on human decisions, emphasis in this research is placed on the influence of both inherent and developed human characteristics per se. which together comprise the ‘individual differences’. It is suggested they have a major impact on decision making and its success. 

Many theories on how business decision making should rationally proceed have been proposed. These range from using mathematical models involving the maximisation of utility, being a measure expressing the total value of an outcome [5] through to an informed use of intuition in achieving the decision maker’s objectives [6]. While the maximisation of utility is often accepted as the theoretically correct approach, in reality few decisions are actually decided in this way. The difficulties involved in being ‘correct’ [7] means many simplifications occur leading through to the use of a decision makers’ intuitive solution in its varied forms [8]. 

In most businesses, particularly small and medium sized operations, decisions are largely made by the manager, sometimes with assistants, using their ‘objectively informed intuition’ [9,10,11] as they believe it provides the best decisions [12], within their knowledge levels, at least cost. Right from the objective set not being the maximisation of utility (the alternative ‘satisficing’ is often mentioned [13,14], as is a ‘lexicographic objective form’ [15], as well as other possibilities) through to the difficulties of quantitatively allowing for risk and uncertainty, the realities mean intuition is a practical and feasible approach even if a decision maker’s position between the practical to the theoretical probably moves with evolving training [16] and experience [17] and similarly for the simplifications used. 

The nature of the decision problem will also influence the position [18] with complex risk laden problems under time pressure leading to the greater use of intuition [3,19,20]. A lack of suitable, and easily interpreted [21], decision data is another factor in using intuition as are the personal depth of thinking attributes of the decision maker [4] which influence the quality of the decisions in complex problems. Overall, as noted by Khatri and Ng [10] (p. 62) ‘intuition is central to all decisions, even those based on the most concrete of hard facts’.

While the literature contains large numbers of articles on intuition (or tacit knowledge) within the business world, there is little consensus over its development and a complete lack of work on quantifying the determinants of intuitive ability. This work contributes by developing a logical, quantified, and consistent understanding of what is being called here ‘objectively informed intuition’ (OII). OII is not a new theory per se, but a descriptive term for the most commonly used approach to business decision making [22] which is sometimes referred to as the dual processing approach [23,24]. More importantly, recognizing the existence of OII leads onto developing a new and unique quantitative understanding of this common decision method. This is essential to improving OII decision skill something that has been lacking in the literature. As business expert Drucker said ‘if you can’t measure it, you can’t improve it’ [25].

The main objective of this work is to demystify the structure of intuitive decision making. This is achieved in the following sections through providing the background to the development of ‘objectively informed intuition’, providing a discussion leading to a quantifiable general theory which is then followed by its testing. To assess the parameters of the theory, data from owner operators of small natural food and fibre producing businesses was gathered to enable specifying, using structural equation modelling (supported by regression analysis), a predictive equation. The business owners, being the managers of small to medium operations, were used as a case situation as they tend to be the main, and often sole, decision maker no matter the number of employees. Finally, a discussion on the methods of improving ‘objectively informed intuition’ and future issues is provided. 

Throughout this study it is recognised that there are important links between decision theory, decision practice, economics and psychology as noted by many researchers, e.g., [26]. Furthermore, aspects of genetics are also important in that gene expression influences decision making and psychology with, in addition, aspects of brain development also playing a part in OII development. Given these influences, various aspects of all these disciplines are introduced with a view to taking an integrated and holistic view of OII including its creation and efficiency. This is the reality of human decision making. 

## 2. Literature on Intuition and Related Issues Leading to the Notion of ‘Objectively Informed Intuition’

While Simon [27] talks about intuition as an analysis frozen into habit, Dane and Pratt [28] encapsulated most of the ideas in the literature with the statement ‘the provision of a conclusion reached without formal analysis’ (p. 40). Sometimes the word ‘heuristics’ (from ancient Greek meaning find or discover) is used in place of intuition. Salas et al. [3] similarly review the concepts but in their case work towards emphasizing decisions made by groups. 

Akinci and Sadler-Smith [29] examined eighty years of management intuition research and refer to humans’ pattern recognition decision making [3,30,31] in which a similar past experience is retrieved and used in the current decision. This process was believed to be an important basis of intuition. In support, Simon [30] estimated a chess grand master holds around 50,000 patterns in long term memory.

However, Akinci and Sadler-Smith [29] suggest the process is more complex involving combining patterns, imagination and cognitive analysis to produce an informed intuitive conclusion for the current decision problem. They also review some of the later MRI scanning work in finding out the brain components engaged under different decision processes referring to the concept of System one (intuition) and System two (formal, logical) decision operations [6]. Glöckner and Witteman [8] extend the idea of dual processing systems and define four types of intuition (associative, matching, accumulative and constructivist) each involving different processes. Other than Akinci and Sadler-Smith [29] none of this work emphasizes ‘pattern matching’. 

While this complex process of ‘objectively informed intuition’ has many additional components, as shown by researchers relating human characteristics to business intuition (reviewed in subsequent sections), researchers have not worked on exploring the core variables defining ‘objectively informed intuition’ and consequently summarizing the influential factors. This work moves in this direction by discovering the relationships between the core variables with a view to understanding the quantitative aspects of decision making in business situations. This understanding is essential to improving business, and other, decision making.

### 2.1. Definition of ‘Objectively Informed Intuition’ (OII)

While intuition is regarded as decisions made without conscious thought [28], where it is possible to have a time lapse [32] between observing details of a situation and the need to act, few would rely on simple immediate intuitive conclusions as is clearly demonstrated by the examples from the literature summarized in Table 1. Furthermore, typical processes used in coming to a reviewed decision, while not directly related to business decisions, are given by Meziani and Cabantous [33] when exploring ‘sense making’ which aptly emphasize the conscious cognitive input commonly used in forming a reviewed conclusion.

Based on the research presented, the common form of business decision making, ‘**objectively informed intuition’** (OII), is defined as: 

the process of coming to a business decision using both initial subconscious conclusions together with modifying mental thought, sometimes involving discussions and research, such that all resources of the mind, both subconscious and conscious, are brought to bear to provide a decision thought by the decision maker to best achieve their objectives.

Salas et al. [3], and others, come to a similar conclusion but refer to OII as ‘expertise based intuition’ being a decision based on joining intuition and expertise. 

The level of active conscious thought in any one decision will depend on the time available for reflection and researching, the nature and complexity of the decision, and the characteristics of the decision maker. The research clearly supports ‘objectively informed intuition’ as the means by which large numbers of decision are based [47] as well as implemented as outlined in the next section. Klein [48] came to a similar conclusion noting 80–90% of decisions are made through a blend of intuition and analysis in the RPD (Recognition-Primed Decision Model) process which is effectively the same as OII.

### 2.2. Managerial Ability, ‘Objectively Informed Intuition’ and Decision Application Skills

Any manager exhibits a level of managerial ability as assessed against the most expert manager for a particular technology and industry (domain). This ability may differ from the manager’s informed intuitive expertise, though it is likely the level of managerial ability and ‘objectively informed intuition’ expertise are similar. To quantify OII it is important this relationship between OII and managerial ability is clarified. 

‘Managerial ability’ normally refers to a manager’s expertise in all attributes required [49] for a particular domain. An example of a necessary skill is formal planning involving, desirably, a knowledge of production economics, markets and future prices and costs over a range of years, and also technical and human relationships knowledge, and similar. Additionally, however, skill in implementing any decision is equally as important and needs to be accounted for in the totality of managerial ability. This application skill is seldom formally recognized in the literature. 

The situation can be expressed through the generalised equation: *Managerial ability* = *a* (*oii* + (*f* × *n*)) (1)
where *oii* = level of objectively informed intuitive (OII) ability

*a* = skill level in implementing an operational plan 

*f* = the decision on whether to use formal analysis, over and above ‘objectively informed intuition’, taking on a value between 1(use) to 0 (not use)

*n* = ability level of manager in formal planning and forecasting

Depending on where the decision maker appears on the continuum from formal decision making through to informal decision making, managerial ability will often reduce to ‘*oii* × *a*’ with *f* = 0, or close to zero, which is clearly a common situation as noted previously with supporting evidence from references.

Both planning ability and the skill level in applying a plan involve many components which need to be included in any modelling as discussed below. However, it must be recognised that outcomes from a manager’s business reflect the totality of their managerial ability so it is not possible to partition ability between decisions on the desirable actions and the skill with which they are implemented. Accordingly it is assumed that the observed business outcomes reflect an OII that involves both the application of intuition in deciding the best plan as well as its implementation. Furthermore, as in most owner/operator businesses little formal planning occurs it is assumed ‘f’ has a value close to 0, or that if f is greater than zero as it will be in a few cases, oii outcomes reflect the totality of the impact of any formal planning, intuition as well as application impacts. The next section brings together the components that are hypothesised to be the core variables of ‘objectively informed intuition’ (OII) involving intelligence as well as other cornerstone variables. 

## 3. Core Variables Influencing Decision Making 

Hodgkinson et al. [37] (p. 8) noted ‘… Literature has lacked a coherent overarching conceptual framework in which to place intuition’. To provide this framework, and to advance a model of ‘objectively informed intuition’, it is necessary to isolate the core variables influencing decision making. 

A review of the literature suggests the core variables most likely to influence OII are (1) a decision maker’s personal characteristics as related to business issues (business related phenotype), (2) the training received, and (3) their experience. Logic would also suggest the level of these three variables is likely to influence the level of OII skill achieved by a business decision maker. The discussion that follows reviews the evidence to support this contention. 

### 3.1. Business Related Phenotype … an Important Building Block

Being the core of behaviour, a decision maker’s personal makeup, in this case, with respect to business issues (which can technically be referred to as their ‘business related phenotype’), including intelligence (involving both genetic and learnt intelligence [50] covering fluid (basic problem solving skill) and crystallised (learnt knowledge and experience) aspects), must significantly influence ‘objectively informed intuition’ based decisions. 

Under the environment experienced each decision maker develops an unique business related phenotype influencing their potential ‘objectively informed intuitive’ ability. In their review, Belsky and Pluess [51] emphasize this noting how different genotypes (the decision maker’s genetic base) will react differently to the environment leading to specific brain development, both physically and cognitively all giving rise to their decision ability (it is thought the process involves variations in the serotonin and cortisol brain chemicals amongst other agents such as dopamine [52]). While the process is likely to be in part probabilistic, it is overall reliant on the concept of ‘brain plasticity’, the ability of the brain to modify its cognitive operation over time in response to stimuli [53].

These factors mean the development of business related phenotype (BRP) is a complex and dynamic system. Furthermore, BRP impacts, for example, on the effectiveness of education and training as well as on how experience provides lessons and knowledge [54]. These relationships emphasize how these three factors will jointly influence ‘objectively informed intuition’. 

### 3.2. Factors in Business Related Phenotype Influencing ‘Objectively Informed Intuition’

As personality and intelligence are core factors defining a person [55] and, thus, their business related phenotype (BRP), they must be included in any model. This is exemplified by the work listed in Table 2 being examples of the many studies showing the relationships.

In addition, a person’s objectives [8,56], while not a direct part of BRP, conduct the cognitive orchestra acting as the mixer of all the components to provide a system best suited to producing the sought after outcomes. As these objectives are unique to each particular decision maker they must also form part of any overall model of ‘objectively informed intuition’ so enabling its assessment.

### 3.3. Training and Interactions with Business Related Phenotype 

Plasticity levels vary with individuals having been influenced by their specific evolutionary progression throughout the ages. Forsman [53] believes plasticity is the determining factor in the phenotypic response to the all important environment and genotype. He talks about active and passive plasticity with, for example, the expression of the shyness phenotype varying depending on the situation. 

Within this situation of BRP and plasticity [6,65] education influences a decision makers’ understanding of the environment in which they eventually operate a well as their communication and personal relationship skills. Important education (training) should also involve an understanding of decision theory and production economics, and other economic theories, even if only at a practical level. A full knowledge of the relevant production technology will also be important to allow decisions taking full cognizance of the input output relationships. Training also includes short courses as well as appropriate reading and discussions with knowledgeable people. Overall, an individual’s decision making will be influenced by the appropriateness and totality of all this learning [66]. 

In addition, each business experience is, potentially, another ‘lesson’ so informal learning will also be part of self regulated activity. BRP will equally influence informal learning through setting the attitude to observation and learning as well as logical thinking. 

Table 3 summarizes examples of the numerous projects relating decision maker characteristics to training achievements including informal learning.

This discussion on training, and Table 3, introduces issues which impinge on a person’s ‘objectively informed intuitive’ skill. It is the essence of the training success, both formal and informal, that leads to improvement in the success of OII based decisions. Consequently training must be an important component of any model. It is also a dynamic issue involving most components of a decision maker’s changing life. 

### 3.4. Experience and Related Skills

Nuthall [49], in researching managerial ability, calculated that experience and the lessons arising accounted for a significant proportion of the factors influencing outcomes. While the specific contribution will vary with business type and eras, experience [81], must be part of any model as demonstrated by Eberhardt et al. [82] who related age and experience to financial decision making success. 

However, to be useful the lessons in each experience need to be isolated, understood and used to positively influence OII. Kolb [83] believes it is necessary to reflect and subsequently develop an abstract conceptualisation of the lesson. This process requires a willingness to learn, an ability to both reflect and conceptualise, and subsequently to make use of the lesson. While this process will be active for many, there will be some cases where benefits occur ‘intuitively’ with non formal learning [84]. 

Many workers have researched experience some of whom are listed in Table 4 as examples. Their conclusions from the work relevant to ‘objectively informed intuition’ is summarized.

Scott [85], Sadler-Smith and Burke [86] as well as Peltie et al. [87] and Matthew and Sternberg [88] concluded active work on reviewing experience and possible lessons was highly beneficial particularly where peer reflection was used. It was generally accepted that recording experiences and formally analysing them with written reflections and conclusions had significant impacts on successful ‘objectively informed intuition’.

### 3.5. Dynamic Development

‘Objectively informed intuitive’ skill has periods of change, with improvement being likely. Any model should recognise this dynamic situation something which few intuition researchers have formally acknowledged. 

Right from birth cognitive skill is changing, and, consequently, so is intuition leading into ‘objectively informed intuition’. This early process is exemplified by the work of Piaget who developed a theory of child cognitive development through to about 20 years of age. Ojose [97] points out Piaget believed there were four basic stages in a child’s development (sensorimotor (finding objects, eye-hand coordination), preoperational (language, symbolic thought and limited logic), concrete operations (using senses to ‘know’ and having two or three dimensions, ordering sequences), and formal operations (developing hypotheses and deducting possible consequences)).

As a decision maker’s life unfolds development will continue in their intuitive skill in general and ‘objectively informed intuition’ (as well as in other areas … [98]) in particular. However, for some managers the four stages, and ongoing development, might not evolve to their ultimate useful levels as exemplified in the range of abilities observed.

There are also additional theories on cognitive development. One of note is Mezirow’s transformational learning theory [99] which proposes that learning transforms and creates cognitive growth. Mezirow believed a person must critically reflect and engage in rational discourse using values, beliefs and assumptions as the lens through which experiences are made sense of. This is an active process with Merriam [99] believing critical reflection is a stage beyond Piaget’s fourth stage with progression through ability stages related to age and education. As a person develops cognitively, they will enhance their ability to make rational decisions relative to their objective set which also changes through life. 

The human characteristics that further develop include abstract thinking, metacognition, relativistic thinking, ‘wisdom’, identity development, concept of self, identity and associated self esteem, environmental recognition, relationship and communication skills, and others, all leading to a unique person with a specific set of beliefs [100,101,102]. Taken as a whole all these evolving dynamic characteristics define a person and influence their potential for successful decision making. 

In putting a decision making framework on these ideas reflecting continuous development, Kelly [103] believed everyone develops ‘personal constructs’ being ‘rules of thumb’ (heuristics) which guide a person’s decisions over all aspects of their life. Through their experience, BRP and training, Kelly [103] noted a decision maker develops an appropriate action for different situations giving rise to their set of constructs. These change with more experience and reflection. Furthermore, encountering a situation for which a construct is not held leads to a developing a new construct. Kelly talked about ‘man the scientist’ in that DMs are constantly reflecting and improving their understanding of the world and its reactions to them through their evolving constructs. Effectively a person’s set of constructs gives rise to their OII through something akin to dynamic pattern matching.

Finally, in fully recognising the dynamic nature of OII, Gegenfurtner and Vauras [104] believe there is evidence that the motivation to learn and transfer knowledge increases with age. Their results also show that learning situations which have a social aspect tend to be more effective and are emotionally meaningful. Kochoian et al. [105] also found other factors were important in motivation including the time until expected retirement, self efficacy, perceived opportunities and their value. These considerations influence the use of short courses in particular and reinforce the need to include age as a component of any model. 

### 3.6. Summary

This assessment of the literature reinforces that the important core variables in the development of ‘objectively informed intuition’ are most likely to be a decision maker’s (1) business related phenotype, (2) training and (3) experience. As experiences are constantly being acquired, and learning can be continuous, OII ability is dynamic with, largely, improvement occurring through time. This change is a result of the totality of a decision maker’s BRP (involving personality, intelligence and the associated observation and anticipation skills) and training combined with the lessons from their experience. Furthermore, a decision maker’s objectives influence the direction of a person’s OII decisions [106] leading to choices that allow, mainly, achieving their objective set. However, as outcomes are uncertain and risky [107,108], it is only on average that good decisions eventuate as they are made before perfect knowledge of all the factors affecting output are known with certainty. 

It must also be recognised many decision makers have decision biases, imperfections, and a lack of knowledge of important influencing factors which all impact on OII success. In addition, given the dynamic nature of ‘objectively informed intuition’, the relative importance of the contributing variables will change with time suggesting the importance of experience and training increase with the relative contribution of BRP declining, though not in absolute terms.

## 4. Quantification of ‘Objectively Informed Intuition’ (OII)

### 4.1. Background

To be useful it is important to develop quantifiable relationships which forecast a decision maker’s OII skill. Popper, an influential philosopher, argued [109] that a central property of science is falsifiability which is facilitated when the hypothesis and corresponding model is quantifiable. Accordingly, an equation which assesses OII skill based on the three core variables was developed. 

The structure of the relationships is expressed in Figure 1. This reflects the logic that correct ‘objectively informed intuition’, the typical decision process, gives high levels of the decision maker’s sought after outcomes (reflected by the box labelled ‘objectives’). That is, optimal ‘objectively informed intuition’ leads to optimal levels of the desired outcomes. Leisure, for example, might be an important objective so an outcome with sufficient leisure is important involving a trade-off between leisure and financial rewards, and additional objectives too.

OII ability is the central variable in the diagram and resultant equation. That this directly unobservable variable is placed in the central position comes from the evidence that OII is the main decision system used by most managers (considerable evidence reflects this as shown in the example references previously quoted). 

The other boxes in Figure 1 reflect that a decision maker’s BRPhenotype, the sum of their training and experience all influence the level of OII skill as concluded from the review of the literature discussed in the previous section (core factors). Formal planning skills, as separate from ‘objectively informed intuition’, is represented in a box reflecting their impact on outcomes as well as influencing ‘objectively informed intuition’. 

The arrows reflect the direction of influence of the inputs with many directions being recognised exemplifying the interactive and dynamic relationships. To quantify the model, data from a group of decision makers is necessary. As it is possible the observations will contain errors, and there may be other factors not yet recognised that may be important, the box at the bottom of Figure 1 reflects the difference between the true model, the hypothesised relationships, and any ‘experimental’ errors.

The need to introduce ‘objectives’ (plural) relates to an inability to objectively measure all the components of a person’s goals using a common scale allowing comparability of values between all experimental subjects. Objective measures of, for example, the pleasure obtained from working in an enthusiastic work force, do not exist for this, and many other, outputs. 

### 4.2. Hypothesis

Given the dynamic nature of ‘objectively informed intuition’, the conclusions from the literature and the logic discussed, as expressed through Figure 1, all leads to the hypothesis that a general equation portraying the OII model is given by:
*oii_t_* = *oii_t_*_−1_ + *Δ f* (*p_t_*, *t_t_*, *e_t_*)(2)
where 

*oii_t_* = decision maker’s ‘objectively informed intuition’ skill level at time *t* (it will be noted that Figure 1 has oii reflecting both planning and plan application skills), 

*p_t_* = individual’s business related phenotype status, at time t, accruing from all previous periods, 

*t_t_* = individual’s past formal training status at time t accruing from earlier time periods, 

*e_t_* = sum of information learnt, at time *t*, from the individual’s past experiences, 

*Δ* = represents change, i.e., the change in *p*, *t* and e over the last period. That is values at time t minus values at time *t −* 1.

For the moment the units of measure are not identified. 

To actually allow the model to be quantified a relationship between output and its precursors is required. Thus: *O(utcome)_j,t_* = *f* (*oii_t_*, *n_t_*, *s_t_*)(3)
where the new variables introduced are…

*n_t_*_=_ formal planning input

*s_t_* = situation of all environmental factors influencing success including prices, costs, resource levels….

*O(utcome)_j,t_* = a variable reflecting the output of each distinct objective *j* at time *t*. 

In the case being proposed it is assumed all decisions are made through ‘objectively informed intuition’ with the assumption that formal documented planning is not directly occurring (which is the main situation), and the situational variable is set at ‘average’ given the range of years available in the data. Thus, *n_t_* and *s_t_* drop from the equation. In this formulation it is assumed *oii* embodies both decision planning and application skills. 

In the hypothesis there are interactions between *p*, *t* and *e* as concluded from the literature. BRP impacts on learning from experience, and training helps learn from experience, and experience influences what training is undertaken though training nor experience directly interact with ‘*p*’, though they might well impact on how well the decision maker understands her/his BRP and accordingly its influence. As noted, an important tripartite exists. 

Due to the measurement comparability difficulties in the objectives, only modified financial measures are used as influenced by relative measures of a range of objectives such as, for example, the rated importance of environmental questions (Figure 1). 

Emotions (affect) at the time of any decision may influence the decision [110,111] being a modifying factor the extent of which will depend on ‘*p*’. Blanchard-Fields et al. [112], for example, found the strategy selected in problem solving was influenced by the ‘emotional salience’ associated with the problem. Sinclair [35] points to the same conclusion, and also comments on feelings when there is contentment with the OII decision. 

However, it is difficult to include affect in any general model with the problems of measuring different emotions at different times relative to any decision conclusion. A dynamic, detailed and contemporary data gathering process would be required. In the proposed model it is assumed with the wide range of data used, and its consistency, affect impinges on an average basis so that for any one decision maker the impact of affect evens out [113]. In any active dynamic model this assumption would need modifying. 

Another dynamic issue is the changing OII skill level with age increases (probably improving). Age needs to be included within ‘*p*’ so partially allowing for the relationship’s dynamic nature. To additionally account for this dynamism it would be necessary to follow a sample of decision makers through several years. 

The measurement unit of ‘objectively informed intuition’ is determined by the units used for the equation variables. They relate to judgements on the strength of belief statements based on a Likert scale including relative measures of variables such as intelligence, personality and age. This leads to the ability to normalize the level of ‘objectively informed intuition’ on a percentage basis comparing all subjects in the sample on a relative basis similar to measuring intelligence (intelligence quotient—IQ). This approach overcomes the lack of an objective scoring system. 

## 5. Obtaining Information from a Sample of Decision Makers to Assess the Hypothesis and Model Parameters

It is not currently possible to objectively measure a decision maker’s mental perceptions and thought processes to enable directly quantifying ‘objectively informed intuition’ [114]. This leads to relying on a sample of decision makers unbiasedly recording their feelings and outcomes [115] (p. 30), [116] based on pre-existing understandings and beliefs [117].

As most natural food and fibre production worldwide is based on family businesses [118,119] it is the individual managers (owner/operators) themselves that make most of the decisions. These small to medium operations are accordingly an uncomplicated ideal base for exploring the nature of ‘objectively informed intuition’. These sampled business manager/owners were questioned in three surveys covering managerial issues provided the data for modelling ‘objectively informed intuition’. The respondents were all randomly selected from full time individual operational managers through sampling from within a national data base (all natural food and fibre producers) containing information on business type, business size and regional strata. The sampling fractions were based on the population proportions. The surveys each provided approximately 650 respondents giving 1940 cases in total. 

The surveys obtained descriptive information on each business (type, land and building details, labour complement …), information on productive levels, and information on each manager such as age, education, grades, experience and similar. The managers were also asked to provide information on their objectives (goals and aims), their managerial style and approaches (business related ‘personality’), and self-grading information. Copies of the questionnaires used in the mail surveys are provided in the on-line reports referred to below. 

Across all businesses the owner/managers averaged 57 years of age (93% > 35) with 34% having tertiary level education, 30% with 4–5 years 2°, and achieved an average grade in the final year of formal education of 63%. The mean debt level in the most recent survey was USD 1,181,800, the capital value USD 4,775,500 giving net assets of USD 3,593,700 with the second survey giving net assets of USD 3,925,800. 

The relevant items of information (81in total) were isolated into an unique data base. Each was either a numeric response (such as, for example, asset investment, labour units, production levels …) or a truth level (1 to 5) for Likert statements. Examples of the statements include ‘it is very important to ensure employees enjoy their jobs’, ‘keeping records on just about everything is very important’, ‘you are much happier if everything is planned well ahead of time’, ‘the years when the business has shown poor production and profit have been due to circumstances totally out of my control’ (the full list of questions and statements in the surveys, as well as factor values and commentaries, is provided in the Appendix A (provided on line at https://researcharchive.lincoln.ac.nz/handle/10182/14465 (accessed on 16 October 2022))).

As the surveys covered a range of periods the monetary information (cash surpluses and asset values changes) was adjusted to a common base allowing for both period and business type. Furthermore, the managers provided five year averages for their output as product prices and situations vary quite significantly from period to period, and from production type to production type. 

For the production information each business was related to the most productive business for each core type to provide a productivity (physical efficiency) coefficient ranging from 0 to 1 (0 to ‘100%’ efficiency). Overall, for each business there was available a measure of their relative cash surplus, their asset value increase, and their relative physical production efficiency. 

Besides the output and managers’ objective information, the survey items were classified according to whether they related to the owner/operators’ business related phenotype, training or experience. Each grouping was factorised to provide a single variable expressing the managers’ levels of BRP, training and experience. The survey information loading onto each variable is given in the Appendix A (available on line at https://researcharchive.lincoln.ac.nz/handle/10182/14465 (accessed on 16 October 2022)) together with the factor loadings. 

Before including each set of data their reliability was checked. In the first set the sample and population strata were compared giving an average discrepancy of 1.21% with a standard deviation of 2.05%. In the second set the Mann–Whitney U test showed there were no significant differences between the business type population and sample statistics. In the final set the Wilcoxon signed rank test similarly showed no significant differences between the response rate population and sample figures (*p* = 0.223) with the Marginal Homogeneity test being *p* = 0.906. For the reliability of the major test, there were no significant differences in a comparison of sub-groups, the Cronbach Alpha showed reasonable reliability and ten randomly selected sub samples of 10% gave a summary value of 66.74% relative to 66.96% for the whole combined sample. 

## 6. Results from the Analyses

As each managers’ objectives are different it is not possible to use simple measures of success such as the cash surplus. Managers with an interest in the environment in contrast to profitability per se would be ranked as poor if only the cash measures were used. Thus, to allow for the varying objectives the answers to the twelve objective statements were factorised using an eigenvalue of one as the cut-off point to the number of factors. The monetary outcomes where then multiplied by the factor values (listed in the Appendix A available ‘on line’ at https://researcharchive.lincoln.ac.nz/handle/10182/14465 (accessed on 16 October 2022)) and summed to provide, firstly, an adjusted physical efficiency measure, secondly, an output using the cash surplus as a base, and a third based on the asset value increases. 

To assess a DM’s OII level (as hypothesised) it is necessary to relate the three dependent measures of adjusted outcomes to the independent variables involving business related phenotype, training and experience obtained through one factor analyses of their constituents. For the BRP the variables recording each manager’s management orientated personality were included as was their age, grade at their last year of formal education, and also the variables making up their attitude to their belief in their ability to control outcomes (LOC) as this reflects their beliefs formed early in their formative years. 

The variables forming the summary variable, through factor analysis, of a decision maker’s training included their education level, their attitude to planning and budgeting skills, their knowledge of management principles, attitude to attending training sessions and similar. For the factor based experience variable the constituents included primarily the years of experience, attitude to seeking help and comment from others over decisions, attitude to checking and double checking the detailed plans made, relying on approaches learnt from experience, and using production methods that have ‘stood the test of time’.

### 6.1. A Structural Equation Model (SEM) Identifying Subjects’ OII Levels as Related to the Explanatory Variables (p,t,e)

While regression analysis provides a knowledge of the importance of the core variables in OII_t_ it does not give their contribution to a manager’s level of ‘objectively informed intuition’ as isolated in Figure 1. ‘Objectively informed intuition’ is not directly observable in the normal sense, but it can be inferred using structural equation modelling (SEM… [120]) which gives the coefficients expressing the importance of the arrows showing the directions of influence. Table 5 presents the relevant parameter values for the SEM solutions.

The comparative fit index [121] compares the model parameters with the assumption that the observed variables are uncorrelated with means zero. Its ideal value is 1.0 being the equivalent of a R^2^ value of 1.0. In this case, the degree of fit is acceptable, and the parameters are all highly significant.

As each owner/manager will have unique ideas on the important outputs it was not logical to use an arbitrary combination of the outputs in forming a single independent variable. Thus the three separate equations reported in Table 5 were estimated.

To relate the core of these results to Figure 1 a simplified version containing the main parameters is presented in Figure 2.

In the SEM the parameters reflect the relationship with the unobservable ‘objectively informed intuition’ which then influences the output variable as a coalition of influences. When the contribution of business related phenotype, training and experience are put into percentage terms the respective figures are 50.7%, 39.1%, and 10.2%. In contrast when simple regression analysis parameter values (see below) are used the contributions to output are 48.9% for BRP, 38.6% for training, and 12.5% for experience. These are remarkably similar to the SEM results confirming the contributions to ‘objectively informed intuition’ with BRP (reflecting in part parental influence) being particularly important.

The results of the SEM model allows estimating each sample member’s ‘objectively informed intuition’ quality level using the parameter values giving rise to OII. Figure 3 gives the distributions for both cash surplus and productivity based ‘objectively informed intuition’ for all sample members (the black lines) as well as groups defined by their level of the core variables business phenotype, training and experience (shaded areas). These were divided into thirds (high, medium and low) giving rise to combinations. A sample of these are presented in Figure 3. The number on each sub distribution represents the combination as defined in the notes below the figure.

It will be noted the ‘all sample’ distributions are normal as would be expected whereas the sub distributions take on a range of shapes. Each covers a sub range of the total sample showing the clear impact of the levels of BRP, training and experience on the outputs each sample member has achieved. It is very clear the higher values of each provide higher quality ‘objectively informed intuition’, particularly for the BRP variable. 

It would have been very useful to have, say, a series of two yearly measurements of all variables for every member of a sample to estimate the dynamic impacts. However, in lieu it was possible to put all managers into three age groups and calculate the various parameters for each. 

This showed the contributions of the core variables to ‘objectively informed intuition’ were, in the case of experience for under 46 year olds, 11%, for 46 to 55 year olds 13%, and for the 56 and older ages 12%. Age clearly causes some impact with an initial increase in the experience contribution. For training the equivalent figures are 21%, 34% and 39% showing a constant increase as would be expected. 

In the case of BRP there is a relative decline of 68%, 53% and 49% with age. With time, training and experience somewhat subsume the contribution of BRP. This is perfectly logical.

The covariance between the independent variables are the same no matter which output is used as they are independent of ‘objectively informed intuition’. They do show the importance of training in interpreting experiences but in the case of BRP there is a negative influence perhaps reflecting biases gathered in early years. When the other interaction terms are included they have small coefficients and are not highly significant. When the ability (planning) variable is similarly added it has a small value and a significance of 0.014, which is surprising. One reason is whether it is indeed measuring the managers’ real planning ability. Furthermore, the CFI drops to 0.39. 

In interpreting the ‘OII to output’ path coefficients it is necessary to consider the units being used. After weighting by the objective factors the cash surplus based variable has a mean of 0.32, the asset value increase based variable a mean of −0.11, and the productivity based variable a mean of 0.04. This explains the different scales of each coefficient given factor scores have means of zero with ranges from negative to positive. 

Overall, the important conclusion is the relative importance of the variables in determining OII. However, it has been noted uncertainty has not been directly included in the relationships as it is difficult to quantify in a survey. If this could have been measured it is expected the level of variance explained would have increased. In the SEM results this lack of a direct measure has not had the same impact as in the regressions reported below being due to the nature of the maximum likelihood calculation used. All the variance has been allowed by individual error variables on the output and cognition variables which are the recipients of all the arrowed directions of influence. It must also be noted affection is not directly measured but appears on an average basis. 

### 6.2. A Linear Regression Equation

For readers not familiar with SEM it is useful to estimate simple linear regression parameters of the hypothesised equation (*O* = *f* (*oii*) = *f* (*p*,*t*,*e*)) enabling reinforcing the SEM results though this analysis does not directly include OII attributing output directly to f (*p*,*t*,*e*). In this process the statistical significance of each variable, including interaction variables, was checked in each of the three regressions with the low significance variables being dropped. Variables with a significance probability of 0.2 or less were retained. As noted, for each of the business related phenotype, training and experience groupings the variables were factorised to produce a single variable for each. The regressions were then repeated using the three weighted objective variables as the independents in various combinations to allow for interactions. This gave the equation parameters shown in Table 6.

It will be noted, however, that the proposed components of OII have little impact on asset increases (R^2^ of 0.12) most likely as land values, the main contributor to asset value increases, are determined by supply and demand with the manager having little, if any, influence. Despite this logic, various combinations of the outputs were experimented with each having a range of differences in parameter values. 

For example, where it is assumed the overall output measure is half cash and half productivity, the R^2^ was 0.23 (0.000) and the business related phenotype standardized coefficient 0.46 (0.000), training 0.01 (0.877) and experience 0.05 (0.044). This, and the other arbitrary combinations, do not add value to the conclusions due, mainly, to their arbitrariness.

For most managers the cash and productivity weighted objective factor outputs will be the most important. As expected the productivity equation explains the highest level of the total variance as a manager has most control over physical output relative to the cash output which is dependent on, in most cases, the prices received on world markets. Half the productivity variance is explained which is again to be expected given the impact of the uncertainty variables such as prices and costs, labour markets, weather and disease as well as other uncertainty variables. Not even the best trained manager’s ‘objectively informed intuition’ can provide perfect forecasts of prices/costs and production conditions. 

Overall, as with the SEM analysis, it is clear the business related phenotype variable dominates followed closely by training leaving experience as being the least influential. This is somewhat surprising though it has to be accepted experience, while valuable, is only of value if interpreted appropriately which in turn depends on a decision maker’s BRP and training.

When it comes to the interactions various combinations were explored with only the significant ones included. Their direct influence is not as great as expected in the total model. However, this does not mean, for example, that training is not important in the interpretation of experience (the pair had r = 0.51 (0.000)), and similarly the manager’s business phenotype’s relationship with training (with r = 0.21 (0.000)). The relationship between BRP and experience was minor and non significant.

In assessing the coefficients only the standardized values are presented as this enables assessing their relative importance. The absolute values by themselves do not have great value as they are not objective measures being based on each managers’ ratings of various issues through the factorisation of each group making up the core variables. 

To explore whether the years in which the surveys were conducted had any influence post the adjustments made, the R^2^ values were calculated for each survey independently. They were much the same for the cash surplus weighted output, but for the asset value weighted change, the first survey had a very low value (0.03 (0.000)) relative to the other two surveys both with a rounded 0.46 (0.000) showing the major impact of era on asset values. For the productivity weighted output the variation was much less (0.35 (0.000), 0.29 (0.000), 0.45 (0.000)) as it only reflects the variation in environmental uncertainty. 

## 7. Discussion

### 7.1. Improving ‘Objectively Informed Intuition’ (OII)

A number of issues are relevant in interpreting the results of the OII analyses including that all three factors (*p*, *t* and *e*) contribute so improving one or more can be worth attention. While Fischer [122] believed cognitive development transcends a range of increasing levels, he also believed it is difficult to assess when a ceiling has been reached. Accepting Fischer [122], a positive approach is to assume work on OII improvement should continue until it appears the rate of change is minimal. This will be impacted in part by age as objectives are likely to change. 

As this current work, using realistic field based business data, clearly shows that *p*, *t* and *e* are all important to ability it is very likely that enhanced training and the efficient utilisation of the lessons from experience can improve OII decision efficiency. Just as importantly, if altering business related phenotype is possible, the gains in OII quality can potentially be considerable. 

For enhancing *p*, *t* and *e* many options are available with individuals likely to have a preferred combination depending on their characteristics [123]. Options include informal learning, formal courses, group work, personal tutors, well structured self learning, and combinations. Kilpatrick and Johns [124] found managers used a range of mixed systems including formal training, discussions with other managers, experts, business organisations, personal observation and experience as well as media material. Furthermore, defining performance goals [3] as targets can encourage and support improvements.

Given the importance of BRP (*p*) a manager must also ask whether important components of BRP can be enhanced using, in particular, cognitive behaviour therapy (CBT) [125]. Conard & Matthews [126] reviewed CBT showing its considerable success, under professional guidance, in business situations. Proudfoot et al. [127], for example, have similarly shown that group session CBT works for improving employee ‘well being’, job satisfaction, productivity, and lowering job turnover. Other studies have also shown how working in groups, and consequently reducing the cost per person, is effective (e.g., [128]). 

Most of the literature accepts training (*t*) is a lifelong process. Sitzmann and Weinhardt [129] discuss what they refer to as ‘training engagement theory’ being a dynamic process involving establishing goals, their priorities, and ensuring persistence in achieving the goals. They believe cognitive ability, self efficacy and an appropriate business training culture are important to success. Accordingly any manager seeking OII improvement should assess likely training requirements for their specific business type and plan a programme accordingly. 

For many, informal learning (ILB—informal learning through observing, questioning, case practice …) is common in improving OII. Cerasoli et al. [130] noted 70% to 90% of work based learning occurs informally being dependent on attitudes, knowledge acquisition and performance, and frequently occurs through work experience being self-initiated, intrinsically driven and self controlled. 

In addition, ILB can occur subconsciously in many cases and is related to an individual’s personality, particularly extraversion, openness and agreeableness. Self efficacy is also related to frequent ILB as is a suitable goal orientation. In an employee situation Cerasoli et al. [130] also note support from supervisors and fellow employees encourages self confidence in using planned ILB.

As shown by Ellis et al. [131], amongst others, experience with failures and successes is a major source of improvement when reflection is used. They comment that the success of systematic reflection, and ‘sense making’ [3] depends on situational factors. Formal reflection involves self explanation (analysis of outcomes considering likely causes), data verification (cross verification) and feedback leading to conclusions. The lessons are enhanced if the decision maker has suitable prior experience, is conscientious and emotionally stable. An ability to evaluate the potential benefits from enhancing performance is helpful. 

Andresen et al. [132] discuss EBL (experienced based learning) and emphasize reflection is important through evaluation and reconstruction using, sometimes, outside assistance. They point out the quality of the reflective thought is more important than the nature of the experience. Debriefing can also be important. Andresen et al. [132] discuss how children can develop their thinking through being encouraged to reflect on their experiences so starting the journey to rational ‘objectively informed intuition’. Furthermore, Quinn et al. [133] emphasize that managers that go through intensive experience, under the guidance of mentors, become noticeably more capable and valuable. Lam et al. [134], however, found the benefits exhibited an inverted U shape outcome depending on a person’s attitude (positive/negative). The skill is in deciding an optimal level.

As noted earlier, some researchers advocate the use of diaries in enhancing ‘objectively informed intuition’ [46,131,135] using the strategies referred to in EBL. This involves recording the background to a decision, the nature and details of the decision, the results stemming from the decision and then a pro-active analysis of the whole experience reflecting on the errors and improvements for enhancing future decisions. 

Many experiments have been carried out to assess whether subconscious based decisions are inferior to combined subconscious-conscious decisions (OII). Many commentators have reported on this work including Nieuwenstein et al. [136], Newell and Shanks [19], Huizenga et al. [137], Bargh [138], and Glöckner and Witteman [8]. Overall, the general conclusion is that it is unclear whether a particular approach is superior. However, as most of the experiments have been laboratory based, often using university students, additional real world work is desirable to allow a practical conclusion. Salas et al. [3] (p. 965), for example, call for ‘more rigorous studies in the field’. This current study clearly moves the conclusion towards OII’s potential superiority (something which Alves et al. [139] also found but in a somewhat different setting). Simple logic similarly follows. However, no matter what the experiments indicate, given most decisions are in reality made through OII, improvement in p, t, and e is likely to be beneficial. 

This review of possible improvement methods does not, however, cover in any detail just what components and aspects of each variable it is most beneficial to alter. Further research is needed to compare and contrast the detailed sub components of *p*, *t*, and *e* thus adding to the results of this work. 

### 7.2. Improvements in the Understanding of OII … Subjects for Further Research

It is clear some DM’s have relatively superior OII. Further work is necessary to better understand the underlying causes of these inherent differences and consequently help in improving an individuals’ OII. Thought needs to be given over the likely underpinning bases of an individual’s OII. These may well include cognitive practice and the related genetic and brain functioning issues, biases, affect and also parental influences over and above simple genetics. 

To aid this work the development and use of OII tests would be helpful. Furthermore, the results of future research along these lines may lead to modifying the impact, and structure, of ‘p’ in the model and possibly lead to adding additional variables. The important categories covering this additional research may well be those listed and discussed in the following sections.

(i)Gene expression and related impacts

There is considerable evidence that recently evolving inheritance concepts may influence business related phenotype, the most important variable (and conceptually may alter how ‘*p*’ is assessed and included in an improved model). Reviewing some of this work leads to the first section in Table 7 which lists examples. The conclusions suggest, due to the influence on a DM of parental, and other forbears, variables reflecting the forbears’ qualities, if available, need assessing for possible influence. This parental effect relates to ensuring the Piaget [140] (and Ojose [97]) stages of brain development are satisfactorily completed resulting in the successful development of cognitive skill. 

These effects relate to the concept of epigenetics where traits can be passed between generations through influencing gene expression in contrast to the genes per se. The parent’s occupations also seem to influence offspring choices and abilities further stressing the potential need to include parental influences in assessing ‘*p*’.

Evidence (see Table 7) also exists showing CBT can have impacts on gene expression permanently influencing BRphenotype. Where a decision maker has undergone management related CBT this should enter the sub-variable measurements influencing ‘*p*’.

(ii)Inconsistency, biases and affectation.

A problem in quantifying OII is man’s inconsistency and the existence of biases [154]. Attitudes and actions can change even given the same situation is repeated, and certainly objectives change with experience and life stage. Detailed recording of affect and its basis will be necessary to unravel some of these impacts which will then allow assessing their impacts on individual’s OII.

The second section of Table 7 lists sample references of research related to affectation and biases. This work suggests emotional processes lead to inconsistency of conclusion some of which will also be biased. This means it is also important to develop tests to assess biases making it clear the source of less than optimal OII for any individual decision maker. Where a particular affectation is consistent, OII skill will be constantly impacted effectively giving a further bias.

Successful tests for these factors may well warrant an additional variable in the OII equation which will likely have a negative coefficient. Such tests will probably be unique to specific domains and different OII skills.

(iii)Brain functions and ‘objectively informed intuition’ (OII).

Research has shown the brain location of most cognitive functions using, mainly, imaging machines. This includes decision making processes. However, this knowledge is, at this stage, of minimal value in modelling OII and assessing individual DM’s OII.

Never-the-less, similar work has shown that many brain functions continue developing throughout life particularly with continued challenging use. This knowledge has OII implications and is one reason for including age in the model. Measuring activities that directly enhance OII may well improve the model’s accuracy and again perhaps warrants including in the training variable when it becomes clearer what needs measuring in this respect.

Similarly, an improved understanding of the OII process itself may further influence an improved model and understanding individual’s possible OII improvement. Figure 4 represents one possibility for the process (Klein [48] reports on a similar, but more complex, model). Assessing such models might involve having a manager making various decisions as brain activity is recorded under a MRI [65] in addition to obtaining what the subject believed was the process.

Evaluating ‘cognitive checking’ in this process model may lead to an additional variable in the model reflecting the decision makers ‘cognitive checking’ ability leading to improved forecasting.

(iv)Testing for OII skill.

As noted, in improving decision processes it would be useful to have a simple measure of OII skill perhaps based on the equation using a questionnaire to determine the key elements of *p*, *t* and *e*.

While not directly testing skill, Allinson and Hayes [163] developed a test to ascertain whether a decision maker tended towards using intuition as against formal analyses. More recently, however, De Bruin et al. [164] worked on developing a test through asking subjects to respond to questions related to potential decision biases. Biases used included resistance to framing, under/over confidence, applying incorrect decision rules and similar. The results were correlated with an index measuring negative life events involving poor decision making. They concluded their test (A-DMC … adult decision making competence) was ‘reliable’. However, the index relied on using variables without objective measures and was compared with life events such as being caught drunk, bankruptcy, and divorce few of which directly relate to business management [165]. Furthermore, it assumes all decision makers have similar objectives which is clearly false unless dealing with corporate businesses with multiple owners where monetary outcomes are the unifying common denominator. In contrast the current work relies on using clearly measurable outcomes modified by the decision makers’ preference rated objectives.

In developing a test the inability to cardinally measure objectives is a significant problem. Furthermore, objectives and cognitive ability are likely to interact in that, for example, initially unobtainable objectives might become available under superior decision skill thus allowing an individual to change their objective set. The measurement difficulty means assessing such an interaction is challenging.

In developing tests it should be recognized that the skill level has probably improved over the centuries with epigenetics and evolving family influences. Certainly knowledge has evolved and is readily accessible in many societies, and educational systems improved all being factors in allowing advanced development of OII. In earlier times where human survival depended more on a decision maker’s inherent skills, families with cognitive systems relatively more suited to OII skills likely prospered with their respective personalities and intelligence.

Collective attitudes have also changed influencing many aspects of ‘objectively informed intuition’ including objectives and attitudes, and consequently the idea of an appropriate decision. Attitudes, for example, to the environment are changing rapidly which all means any test must be constantly changing.

(v)Implications and limitations

The issues raised in this section show there are many aspects to be researched further. This work must involve systems thinking [166,167] as especially in ‘objectively informed intuition’ it is important to allow for interconnected factors in a complex whole. Counter currently, Jung [1] believes intuition is necessary for holistic thinking suggesting all, particularly researchers and thinkers, must train their ‘objectively informed intuition’ accordingly. Thinking holistically will also be important in allowing for ‘affect’. Equally, system thinking will indirectly include feedback and related issues.

This study uses managers’ answers to questionnaires, and while statistically it is possible to conclude on the relative importance of the core variables, it does not categorically prove they determine OII levels. To overcome this common problem it would be necessary to use controlled experiments using realistic business decision situations, an approach both practically and ethically difficult. The conclusion is that statistical studies must currently form the main basis of business decision situations research.

It must also be noted the parameters of the equation relate to a specific domain. It is possible they will vary with domain a conclusion which must wait for further research. It is suspected the relative importance of each variable (*p*, *t* and *e*) will not vary greatly but such a conclusion must wait for further work.

## 8. Conclusions

Considering the analysis, which operated under the logic of OII being the most important and common decision process, it showed the relative importance of *p*, *t* and *e* (in both sets of outcome data …production efficiency and cash output) as components of the hyothesised OII equation.

Averaging the analysis sets gives the importance of each variable in contributing to decision success with BRP being at 50%, training at 38% and experience at 12%. If all the SEM and regression results are averaged the figures are not that different with BRP at 50%, training at 39% and experience at 11% (a small shift downgrading experience somewhat).

It is often said that some managers are born with good intuition, and others not. While this is certainly true, it does not mean their intuitive skill is fixed for life. The results of this work, both the review and analyses, show this is certainly incorrect. This provides hope for all managers in that they can improve particularly through using the improvement techniques reviewed and discussed.

The age analysis shows OII improves, on average, throughout life showing the dynamic nature of OII. There is also evidence that having supportive parents with respect to decision making improves OII [49]. This OII improvement is also clearly in the hands of the DM through a decision to work extensively on its improvement. It should also be noted that more than likely core intuition (System one [6]) improves over time as the use of OII and System two (analysis [6]) gives conclusions that then permanently influence core intuition expressing the dynamic nature of OII.

It should also be remembered that any DM faced with a decision problem, even given the core variable equation estimate knowledge, must assess whether they should rely on OII relative to a more formal decision making approach (System two). In a subjective sense, this must depend on the quality of their ‘objectively informed intuition’ in each decision problem something which the equation can help predict.

In areas where the decision maker has little experience, where unfathomable complexity exists but considerable formal data is available, where forecasts are available and likely to be accurate, the probability of a formal analysis being superior is enhanced. However, a DM may not have the knowledge for a formal analysis, and some will not be bothered. For further discussion on this conundrum, Glöckner and Witteman [8] is a valuable reference.

Decision makers likely pass through various levels of OII excellence as they gain experience, perhaps subconsciously in some cases (much like Kelly’s [103] ideas of ‘man the scientist’). Completed formal improvement processes will influence the current stage of the development. While this is probably continuous, stages may well be identifiable.

Initially ‘objectively informed intuition’ will be more like simple intuition so it might be called

(i)‘intrinsic and uninformed’, then moving onto(ii)‘experienced informed subconscious’, next(iii)‘training informed’ leading to(iv)‘cognitive influenced’ and finally a decision maker can be classed an(v)‘OII expert’.

The various stages are not likely to be linear, but dynamic and piecemeal. More research will be necessary to explore the development process and stages. Furthermore, OII may well peak much like face recognition skills peaking at 30 years following an inverted U shape graph [160].

This work does not consider combining decision makers’ OII in a multi individual decision maker environment [3] typical for group decisions in the corporate world. To achieve this the model will need to have sections added that allow for personal relationships and group conclusions [33,47,168,169,170,171]. It is suggested this would add considerable complexity with little previous work to draw on.

Overall, the results of this analysis provide clear direction...

**for an individual to improve their OII, they must work on the methods that cognitively modify their knowledge, attitudes and approaches to make them as close as possible to a personality and knowledge level** [172,173] **of the known best decision makers, that is to modify ‘*p*’ and improve ‘*t*’ and ‘*e*’.**

As good a set of outcomes as possible under the current circumstances will then eventuate. In this process it is likely imagination and inspiration will be necessary to go beyond the past both in terms of improving OII skill and of the skill level itself. Imagination is partially genetic and partially learnt creating a further challenge. It was no less than Einstein who said imagination was the greatest of all human assets [174].

Finally, it is useful to note Sinclair [35] concluded (p. 384) ‘More than ever, there is a need to develop a comprehensive intuition model that should refute some of the misconceptions’. This work has uniquely and significantly moved in this direction for the business decision making world with this research defining and exploring the variables giving ‘objectively informed intuition’ ability.

The results make it clear the hypothesis proposed is a critical and significant part of understanding intuition and OII. Subsequent research must further develop the conclusions particularly with respect to the dynamic aspects which will require repeated surveys of the same subjects over many years. There are many challenges for enhancing this new understanding of ‘objectively informed intuition’ consequently increasing the potential benefits to business decision makers in particular, but also more generally.

As OII is a common human decision process, it deserves an unique descriptive name. ‘Cognuition’ is possibly that word.

## Figures and Tables

**Figure 1 behavsci-12-00409-f001:**
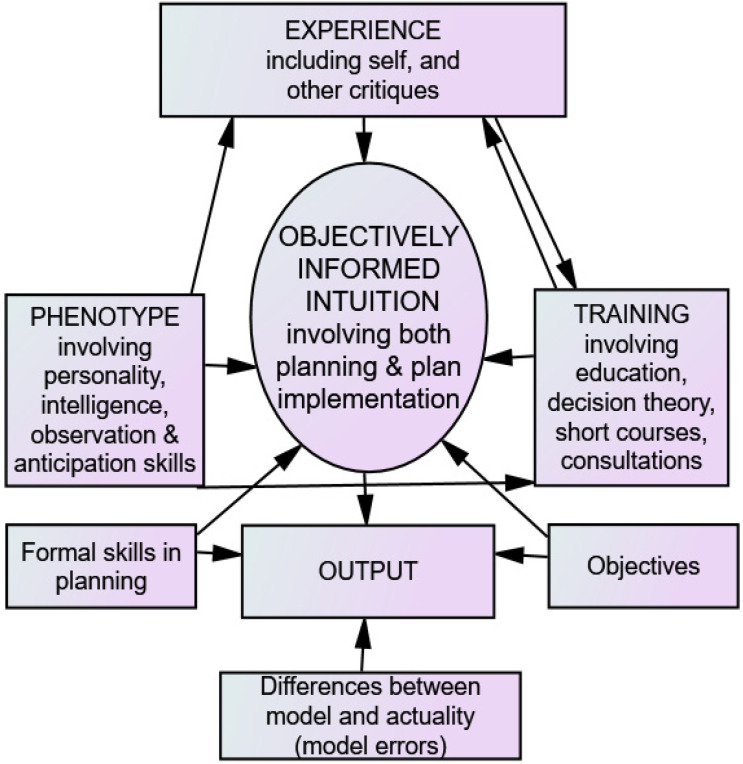
An outline of a model explaining the contributors to ‘objectively informed intuition’ decision making and the resultant output.

**Figure 2 behavsci-12-00409-f002:**
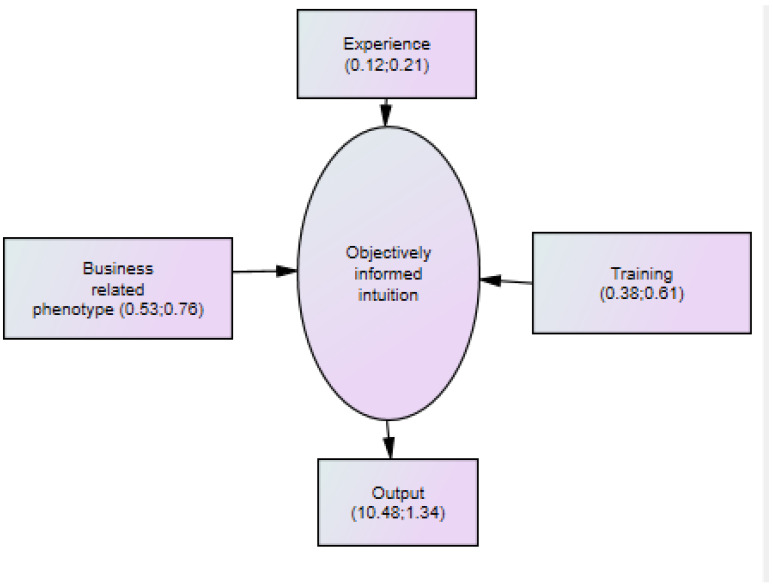
A simplified version of the SEM model showing the main results from the analysis. The model solution coefficients are shown in the brackets the first being for the objective ‘surplus cash’ weighted by the objective factors, and the second for ‘physical productivity’ similarly weighted by the objective factors. Each coefficient represents the parameter of the arrow representing the relationship between the input variable and the output (either weighted cash or physical productivity). The third relationship using the asset based objective is not shown being less important. The goodness of fit/reliabilty for the first objective (the others are similar) is given by CFI of 0.9, the NFI of 0.9, the RMSEA of 0.1 and the Hoelter of 148. They relate to the vary large sample of 1940).

**Figure 3 behavsci-12-00409-f003:**
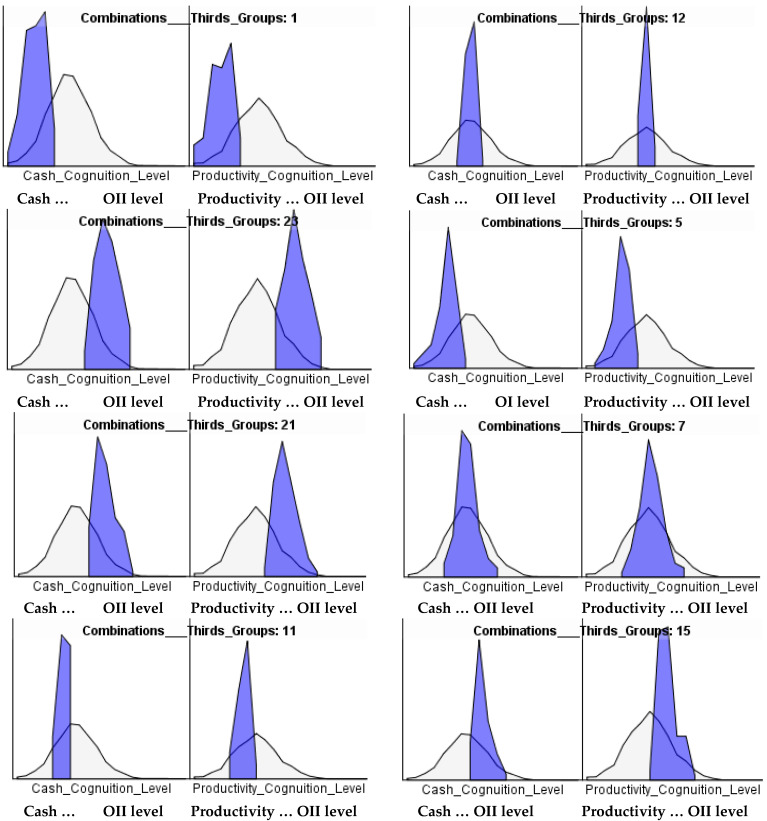
Frequency distributions of both the cash based and productivity based ‘objectively informed intuition’ (OII) variables (clear black line curves) relative to their distributions (shaded) for ranges of the BRphenotype, training and experience variables divided into thirds (the number on each sub graph reflects the combinations). For each group of the BRphenotype, training and experience variable levels, respectively: 1 = low, low, low combination; 12 = medium, medium, medium combination; 23 = high, high, high combination; 5 = low, low, high combination; 21 = high, high, medium combination; 7 = low, high, high combination; 11 = medium, low, medium combination; 15 = medium, high, high combination.

**Figure 4 behavsci-12-00409-f004:**
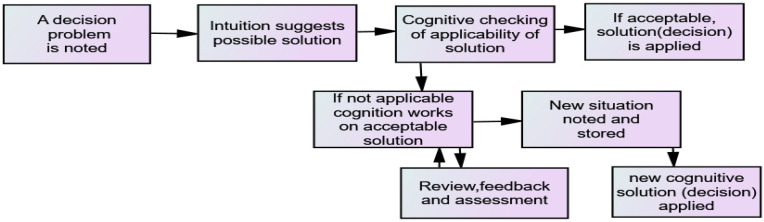
Possible pathways in the OII process.

**Table 1 behavsci-12-00409-t001:** Examples of the literature supporting the concept that most intuitive decisions rely on a conscious cognitive input and review where there is a time lapse between observation and the need to act.

Researcher/s	Findings Relevant to This Study
Aczel et al. [34] and Sinclair [35].	Lack of awareness of decision processes ended in poorer performance suggesting cognition is helpful. And suggests ‘intuition blended with deliberation’ occurs.
Baldacchino et al. [36], Salas et al. [3] and Hodgkinson et al. [37].	Outlines a dual process theory of both unconscious and deliberative processes which is supported by neuroscience (Lieberman [38]). Also ‘affect’ has an influence.
Glockner & Witteman [8], Horstmann et al. [39],	Deliberation is involved in instruction induced decisions, and generally involves deliberation. Also ‘affect’ influences decisions.
George & Dane [40] and Fredrickson & Branigan [41].	Positive attitude influences cognitive responsiveness.
Juanchich et al. [42].	Personality and decision style influences intuition.
Baumeister et al. [43].	Conscious thoughts influence behaviour. ‘(create) behavior (that) emerges from a combination of conscious and unconscious processes’ and ‘allows behavior to be informed by social and cultural factors’ (p. 353)
Pretz et al. [44].	Showed that ‘inferential’ intuition, based on analytical processes, is distinct and correlated with good judgement showing the importance of analysis.
Walker et al. [45].	Believe decision speed influences outcome thus encouraging taking time to reflect.
Hodgkinson & Healey [46], Pretz et al. [44] and George & Dane [40]	All note the importance of ‘affect’ in decisions suggesting cognitive influence.

**Table 2 behavsci-12-00409-t002:** Examples of research of phenotypic influences on cognitive intuition (‘objectively informed intuition’).

Researcher/s	Findings Relevant to This Study
Volkova & Rusalov [57].	Relationships between personality and cognitive styles are significant [58].
Dewberry et al. [59].	Personality explains significant variance in decision competence (p. 787).
Pretz et al. [44].	Aspects of personality influences type of intuition used. Also, with intelligence, influences looking ahead skills [60].
Glen et al. [61]; Sekaran & Bougie [62].	Personality and intelligence influence observation skills as the precursor to planning.
Hodkinson & Healey (2011) [44].	Influences on implementation skills, personal relationships and labour motivation.
Gaglio [63]; Sinclair [35] & Baldacchino et al. [36]. Hough & Ogilvie [64].	This group note the impact of phenotypic components, and training and experience, on entrepreneurship, original thought, creative intuition, and new ideas and their analysis. Cognitive style as related to personality influences strategic decision outcomes.

**Table 3 behavsci-12-00409-t003:** Examples of research of phenotypic and other influences on training.

Researcher/s	Findings Relevant to This Study
Naquin & Holton [67].	Personality (conscientiousness and agreeableness) related to training motivation, and extraversion explained 57% of variance.
Esfandagheh et al. [68].	Significant relationships between personality (extraversion and conscientiousness), and the Locus of Control [69], with training outcomes and speed, and motivation to learn.
Roberts et al. [70].	Locus of Control mediated control of learning, and ‘proactive personality’ was related to leaning motivation and success.
Rowold [71].	Extraversion and agreeableness (personality) were important to learning success.
Bidjerano & Yun Dai [72].	Five factor personality model [73] defined components of self regulated learning including critical thinking and time management…., and GPA correlated with model components other than neuroticism (lowers GPA).
Noe et al. [74].	Found ‘zest’ (energy, anticipation….) correlated with informal learning.
Komarraju et al. [75].	Found elaborative processing, methodical study and fact retention highly correlated with personality components.
Smith et al. [76] and Bell & Ford [77].	Discovered that the decision maker’s goals and motivation was related to training success.
Ainley [78], Efklides [79], and Jakešová & Kalenda [80]	Explored societal and parental influences on learning through emotions developed and their impact on learning. Bad, relative to good, emotions impacted on outcomes.

**Table 4 behavsci-12-00409-t004:** Examples of research on experiential influences on ‘objectively informed intuition’.

Researcher/s	Findings Relevant to This Study
Hogarth [89] and Salas et al. [3]. Nuthall [90].	Lessons from experience requires accurate observation. Feedback is valuable helping memory storage and ideas on modifications of lessons. Concepts need repeating three times before understanding occurs (on ave.).
Shanteau & Stewart [91] and Plessner et al. [92].	Feedback must be timely and accurate for experience lessons to be useful and used sensibly in updating OI rules [93].
Matthew & Sternberg [88].	Critical thinking is important for experiential lessons to embed in ‘objectively informed intuition’.
Cox [94] and Eraut [84].	Structured reflection is important in purposeful lessons from experience and also helps in memory embedding.
Zhao [95]. Vera et al. [96].	Judgement errors lead to emotions strengthening the desire to improve learning from experience. Reflection with personality, conscientiousness and emotional stability helps as does an awareness of societal attitudes. Factors associated with thoughtful decision making related to lack of experience.

**Table 5 behavsci-12-00409-t005:** The SEM solutions to the base form of Figure 1 expressing the importance of the core variables and their relationship with ‘objectively informed intuition’.

Output Type → Variables	Objective Factor ‘Surplus Cash’ Weighted	Objective Factor ‘Asset Increase’ Weighted	Objective Factor ‘Physical Productivity’ Weighted	Average Coefficient
BRPhenotype -> ‘objectively informed intuition’	0.53 (0.000)	0.25 (0.000)	0.76 (0.000)	0.51
Training -> ‘objectively informed intuition’	0.38 (0.000)	0.19 (0.000)	0.61 (0.000)	0.39
Experience ->’objectively informed intuition’	0.12 (0.000)	0.08 (0.000)	0.21 (0.000)	0.10
‘Objectively informed intuition’ -> Output	10.48 (0.000)	38.17 (0.000)	1.34 (0.000)	16.66
Training <-> experience	0.36 (0.000)	0.36 (0.000)	0.36 (0.000)	0.36
Experience <-> BRphenotype.	−0.20 (0.000)	−0.20 (0.000)	−0.20 (0.000)	−0.20
Comparative fit index (CFI)	0.93	0.86	0.94	

**Table 6 behavsci-12-00409-t006:** The standardized regression parameters for each of the objective modified outputs, together with their significance levels (in brackets).

Output Type → Variable	Objective Factor ‘Surplus Cash’ Weighted	Objective Factor ‘Asset Increase’ Weighted	Objective Factor ‘Physical Productivity’ Weighted	Average Coefficient
BRPhenotype	0.40 (0.000)	0.24 (0.000)	0.45 (0.000)	0.36
Training	0.31 (0.000)	0.18 (0.000)	0.37 (0.000)	0.29
Experience	0.09 (0.000)	0.07 (0.002)	0.11 (0.000)	0.09
BRPhenotype × training	−0.03 (0.090)			
BRPhenotype squared	0.10 (0.000)			
Training × experience		0.01 (0.556)		
BRPhenotype × experience			0.05 (0.003)	
BRPhenotype × experience squared			0.06 (0.001)	
R^2^	0.36 (0.000)	0.12 (0.000)	0.44 (0.000)	

**Table 7 behavsci-12-00409-t007:** Summary of research on the possible refinement of the parameters and their values in the model of OII (covering … Gene expression and related impacts; Inconsistency, biases and affectation; Brain functions and ‘objectively informed intuition’).

Researcher/s	Findings Relevant to This Study
Braun & Champagne [141]; Willbanks et al. [142]; Ballestar [143] Bechara & Damasio [144]; Feinstein & Church [145]; Yehuda et al. [146]; Heim & Binder [147]. Pablo-Lerchundi et al. [148]; Oren et al. [149]. Epstein [150]; Casey et al. [151]; Ojose [97]; Piaget [140]; Gormley et al. [152]; Cruce et al. [153]. Nuthall [154]; Damasio (Clark et al. [155]); Alós-Ferrer and Hügelschäfer [156]; Hilbert [157]. Burns [158]; Hilbert [157]; Hodgkinson et al. [37]; Alós-Ferrer and Hügelschäfer [156]; Armstrong [159]; Zhang [160]; Salas et al. [3]. Armstrong [159]; Sinclair [35]; Nelson [161]; Germine et al. [162]; McElroy et al. [4].	**Emerging concept of epigenetics and gene related impacts** The benefits gained by parents through their experience can potentially be passed onto offspring through what has been termed ‘epigenetics’. Environmental behaviourally induced effects can be passed on to offspring without DNA change…mice experiments, human identical twin studies. Enhanced ability to imagine future outcomes from the development of the prefrontal brain. Psychotherapy shifts patterns of gene expression providing permanent benefits which epigenetically can be passed on. Children tend to follow parent’s profession possibly through epigenetic and direct environmental impacts. Brain development and Piaget’s stages with possible impairment and enhancement from parental influences leading to eventual choice of activity and OI development. Superior abilities in family line. **Inconsistency, biases and affectation** Most decision makers have biases influencing decisions inappropriately. Biases need identifying and altering. Emotion (affect) leads to inconsistency of biases. Various tests developed to effectively assess biases and cognitive distortion. Most use inappropriate measures for business situations. Decision types should influence tests. Expertise is domain restricted and processes change with more complexity in decision problem all of which should be allowed for. **Brain functions and ‘objectively informed intuition’ (OII)** Studies have considered where OII activity is positioned in the brain but without influencing models. Never-the-less, also relates to brain maturation and improvement with age and experience all related to stimulation and practice. However, some aspects decline with age.

## Data Availability

The data used is not publicly available due to the confidentiality agreement made with the subjects. However, in special cases the data can be provided, and summaries and tables are available on request on the WWW.

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
