# Peer review of "Assessing the Core Variables of Business Managers’ Intuitive Decision Ability: A Review and Analysis"

_behavsci, 2022, doi:10.3390/bs12110409_

Round 1
Reviewer 1 Report
The article is interesting, the researched problem has scientific huge potential, and the literature review represents a strength. However, three problems need to be solved:
1. For the SEM analysis, please present the reliability and reliability measures of the model as well as the fitting measures.
2. A graphic representation of the model obtained after SEM would increase the paper's intelligibility.
3. In my opinion, a section of conclusions that includes theoretical and managerial implications, research limitations, and future research directions would be helpful. The current form of the conclusions briefly repeats what was stated in the results and discussion section.
Reviewer 2 Report
The authors have made a decent attempt at exploring the contribution of intuitive decision ability.
The following changes will refine the structure and depth of the paper:
1) The abstract can be made more reader-friendly by reducing the elaboration of findings and instead increasing the focus on theoretical and practical contributions.
2) The introduction section can be tuned better with a clear focus on research objectives. More recent references can be valuable.
3) The literature review is a good read and detailed. However, it can be made more interesting with an argumentative approach rather than descriptive.
4) A new section of study limitations can be helpful.
5) The section on implications is weak. It should be rewritten by connecting study findings with theoretical and practical contributions.
6) A new section on future research can be added towards the end for those who are interested in extending the findings from this paper.
Reviewer 3 Report
Thank you for the opportunity to read this paper. The subject is both interesting and engaging.
The title of the paper reflects the main idea, my only concern would be regarding the way the "..." fith in the title.
The abstract of the paper comprises the main ideas and findings, while the introduction and the theoretical background analysis successfully present the findings in literature.
Overall, the appreciations regarding the research, the presentation of the main findings and the Discussion and Conclusions part are positive.
